# Quantitative imaging reveals real-time Pou5f3–Nanog complexes driving dorsoventral mesendoderm patterning in zebrafish

Mireia Perez-Camps[1]*, Jing Tian[1†], Serene C Chng[1], Kai Pin Sem[1], Thankiah Sudhaharan[1], Cathleen Teh[2], Malte Wachsmuth[3‡], Vladimir Korzh[2,4]*, Sohail Ahmed[1]*, Bruno Reversade[1,2]*

[1]Institute of Medical Biology, A*STAR, Singapore, Singapore; [2]Institute of Molecular and Cell Biology, A*STAR, Singapore, Singapore; [3]Cell Biology and Biophysics Unit, European Molecular Biology Laboratory, Heidelberg, Germany; [4]Department of Biological Sciences, National University of Singapore, Singapore, Singapore

**\*For correspondence:**
perezcamps.m@gmail.com (MP-C); vlad@imcb.a-star.edu.sg (VK); sohail.ahmed@imb.a-star.edu.sg (SA); bruno@reversade.com (BR)

**Present address:** [†]College of Life Sciences, Northwest University, Xi'an, China; [‡]Luxendo GmbH, Heidelberg, Germany

**Competing interests:** The authors declare that no competing interests exist.

**Abstract** Formation of the three embryonic germ layers is a fundamental developmental process that initiates differentiation. How the zebrafish pluripotency factor Pou5f3 (homologous to mammalian Oct4) drives lineage commitment is unclear. Here, we introduce fluorescence lifetime imaging microscopy and fluorescence correlation spectroscopy to assess the formation of Pou5f3 complexes with other transcription factors in real-time in gastrulating zebrafish embryos. We show, at single-cell resolution in vivo, that Pou5f3 complexes with Nanog to pattern mesendoderm differentiation at the blastula stage. Later, during gastrulation, Sox32 restricts Pou5f3–Nanog complexes to the ventrolateral mesendoderm by binding Pou5f3 or Nanog in prospective dorsal endoderm. In the ventrolateral endoderm, the Elabela / Aplnr pathway limits Sox32 levels, allowing the formation of Pou5f3–Nanog complexes and the activation of downstream BMP signaling. This quantitative model shows that a balance in the spatiotemporal distribution of Pou5f3–Nanog complexes, modulated by Sox32, regulates mesendoderm specification along the dorsoventral axis.

## Introduction

Oct4 is a key transcription factor (TF) in the pluripotency regulatory network (*Boyer et al., 2005*; *Esch et al., 2013*; *Loh et al., 2006*; *Pan et al., 2002*). Previous studies have suggested that Oct4 drives cell lineage commitment in a dose-dependent manner (*Niwa et al., 2000*; *Thomson et al., 2011*), switching gene regulatory regions (*Aksoy et al., 2013*) by diffusive behaviors (*Kaur et al., 2013*; *Plachta et al., 2011*) and mediating changes in chromatin structure (*Abboud et al., 2015*; *Carey et al., 2014*; *Hogan et al., 2015*). The mechanism by which Oct4 is able to specify naive cells in a pluripotent state into one of the three germ layers is not completely understood. The function of TFs can be modulated by their interactions with other TFs (*Phair et al., 2004*). In vitro assays of Oct4 protein interactions have been performed (*Liang et al., 2008*; *van den Berg et al., 2010*; *Wang et al., 2006*); however, they do not reflect the complexity of a live embryo and do not provide information about the spatiotemporal complex formation that occurs during embryonic development. Therefore, to determine the mechanism by which Oct4 regulates cell fate decisions and patterning, it is important to investigate Oct4 complex formation with other TFs in the context of a developing embryo.

**eLife digest** As an animal embryo develops, cells divide and establish three distinct layers called the ectoderm, mesoderm and endoderm. Proteins called transcription factors control this process by regulating the activity of particular genes. Two or more transcription factors may interact to modulate each other's activity. Zebrafish embryos provide an ideal model system for monitoring how these embryonic layers form and the interactions between transcription factors in real-time because they are transparent and develop outside their parents. Pou5f3 and Nanog are two key transcription factors involved in this process in zebrafish. However, it is not clear how Pou5f3 and Nanog instruct cells to become ectoderm, mesoderm or endoderm.

Perez Camps et al. used imaging techniques to study Pou5f3 and Nanog. The experiments show that Pou5f3 and Nanog bind together to form complexes that instruct cells to form the temporary layer that later gives rise to both the mesoderm and endoderm. The cells in which there are less Pou5f3 and Nanog complexes form the ectoderm layer.

To develop the body shape of adult zebrafish, the embryos need to give individual cells information about their location in the body. For example, a signal protein called bone morphogenetic protein (BMP) accumulates on the side of the embryo that will become the underside of the fish. Perez Camps et al. show that once the endoderm, mesoderm and ectoderm have formed, Pou5f3–Nanog complexes regulate BMP signalling to specify the underside of the fish. Meanwhile, in the endoderm on the opposite side, another transcription factor called Sox32 binds to individual Pou5f3 and Nanog proteins. This prevents Pou5f3 and Nanog from forming complexes and determines which side of the embryo will make the topside of the fish. A future challenge is to explore other transcription factors that may prevent Pou5f1 and Nanog from binding in the mesoderm and ectoderm of the topside of the fish.

Mammalian Pou5f1/Oct4 can functionally replace its paralogue Pou5f3 (*Frankenberg et al., 2014*) in zebrafish embryos, with evidence showing that overexpressed Oct4 can rescue the phenotype of maternal-zygotic (MZ) *spg* embryos that lack Pou5f3 function (*Onichtchouk et al., 2010*). Maternal Pou5f3 regulates dorsoventral (DV) patterning (*Belting et al., 2011*; *Reim and Brand, 2006*) and endoderm formation (*Lunde et al., 2004*; *Reim et al., 2004*), whereas the establishment of the mid–hindbrain boundary requires zygotic expression of Pou5f3 (*Belting et al., 2001*; *Reim and Brand, 2002*). Pou5f3 induces mesendoderm ventralization through activation of the BMP pathway and the expression of the Vent (Vent, Ved and Vox) family of TFs (*Reim and Brand, 2006*). In addition, although Pou5f3 is required in mesendoderm progenitors for *sox17* activation to specify endoderm formation (*Lunde et al., 2004*; *Reim et al., 2004*), it is not required for upstream regulators of endoderm, which are properly induced in MZ*spg* (*Lunde et al 2004*; *Reim et al., 2004*). In contrast, Nanog is critical for endoderm induction through the Mxtx-Nodal pathway where it induces *sox32* in mesendodermal cells and other early, endoderm regulators, such as *gata5, mixer, nrd1* and *nrd2* (*Xu et al., 2012*). Uniquely, Sox32, in the presence of Pou5f3, activates *sox17* expression in endodermal cells (*Alexander et al., 1999*; *Kikuchi et al., 2001*; *Lunde et al., 2004*; *Reim et al., 2004*).

Loss- and gain-of-function genetics experiments, as well as investigations at the mRNA level, have sought to identify various roles for Pou5f3 (*Belting et al., 2001*; *Burgess et al., 2002*; *Lunde et al., 2004*; *Onichtchouk et al., 2010*; *Reim and Brand, 2006*), Nanog (*Schuff et al., 2012*; *Xu et al., 2012*) and Sox32 (*Kikuchi et al., 2001*; *Reim et al., 2004*) during zebrafish development. Here, we exploit fluorescence lifetime imaging microscopy (FLIM) and fluorescence correlation spectroscopy (FCS) to study, at the protein level, the TF complexes and dynamics that underlie cell fate commitment in vivo. We present a quantitative model to describe how Pou5f3–Nanog complexes, modulated by Sox32, can specify mesendoderm cell lineage differentiation in a spatiotemporal manner along the DV axis.

## Results

### Pou5f3-bound active fraction regulates early zebrafish development

To investigate how Pou5f3 controls early cell lineage differentiation in vivo, we used a phenotype complementation assay to rescue MZ *spg* mutant embryos with a GFP-Oct4 fusion protein. The GFP-Oct4 fusion protein was able to complement the *spg* phenotype in 30% of injected embryos. Because rescued embryos could only be identified from 75% epiboly onwards, we could not analyze earlier developmental events. Alternatively, morpholino (MO)-mediated knockdown of maternal Pou5f3 specifically blocks Pou5f3 activity in 100% of injected embryos, which arrest at the blastula stage (*Burgess et al., 2002*). This depletion approach allowed us to discriminate embryos that are rescued by the *GFP-Oct4* mRNA from those not rescued, which remained arrested at the blastula stage. *Figure 1—figure supplement 1* shows the phenotypes of *pou5f3* morphants and the details of the rescue.

FCS (*Figure 1a*) has been previously used to study dynamic processes, such as blood flow (*Pan et al., 2007*) or morphogen gradients (*Yu et al., 2009*) in living zebrafish embryos. Recent studies have described the use of FCS to analyze TF protein activity in iPScells (*Lam et al., 2012*) and pre-gastrula mouse embryos (*Kaur et al., 2013*). In cells, TFs can be found free (free fraction, $F_1$) or as complexes poised to interact with DNA and regulate gene expression (bound fraction, $F_2$). Using FCS (*Figure 1a*), we hence sought to ascertain fluctuations in the fluorescence intensity of GFP-Oct4 over a timeframe of milliseconds and calculate the autocorrelation functions (ACFs) at different developmental stages. To obtain GFP-Oct4 protein concentrations and diffusion kinetics of single cells in rescued embryos, the ACFs were fit using the two-component anomalous diffusion model (Material and methods and *Figure 1—figure supplement 2*). At the blastula stage (oblong stage; 3.5 hpf), the GFP-Oct4 concentration was $44.39 \pm 1.54$ nM (*Figure 1c–e* and *Figure 1—source data 1*). Non-rescued embryos arrested at the oblong stage (*Figure 1b*) had similar Oct4 concentrations ($43.90 \pm 18.30$ nM) to those of rescued embryos (*Figure 1c–e* and *Figure 1—source data 1*). However, the DNA-bound fraction ($F_2$) was significantly lower in the non-rescued embryos ($0.19 \pm 0.08$) as compared with the rescued ones ($0.27 \pm 0.01$) (p<0.0001; *Figure 1c–e* and *Figure 1—source data 1*). When a construct of Oct4 lacking its homeodomain (GFP-Oct4ΔHD) was used instead, the DNA-bound fraction further decreased, and all embryos arrested at the oblong stage ($0.12 \pm 0.01$) (p<0.0001; *Figure 1c–e* and *Figure 1—source data 1*). These results suggest that specific levels of the Oct4 DNA-bound active fraction—and, by extension, Pou5f3—are crucial for proper embryo gastrulation.

### The TFs Pou5f3 and Nanog complex in vivo in the mesendoderm lineage

We next set out to determine the tissue-specific distribution of the DNA-bound active fraction ($F_2$) of Pou5f3 within a developing embryo at the blastula stage (oblong; 3.5 hpf) using GFP-Oct4. In zebrafish, unlike in mammals, it is possible to follow mesendoderm and ectoderm formation during early development (*Fong et al., 2005*) (*Figure 2a* and *Figure 2—figure supplement 1*). For that purpose, zygotes were injected with *GFP-Oct4* mRNA followed by dextran red as a lineage tracer into two central cells at the 16-cell stage, such that the dextran red-positive cells are entirely ectodermal, whereas 83.7% of the negative cells are mesendodermal (*Figure 2—figure supplement 1*). FCS measurements were performed for both labeled and unlabeled cells. The ACFs of the intensity traces were fit using a two-component anomalous diffusion model. Mesendodermal cells showed a significantly higher proportion of the DNA-bound active fraction of GFP-Oct4 protein as compared with ectodermal cells ($0.27 \pm 0.01$ *vs* $0.19 \pm 0.01$, respectively; p<0.0001; *Figure 2—figure supplement 1* and *Figure 2—source data 1*). Dextran red-labeled mesendodermal cells were used as a control for the assay (*Figure 2—figure supplement 2*). Thus, at the oblong stage, the DNA-bound active fraction of Oct4 was higher in the mesendoderm than in the ectoderm lineage.

Next, the GFP-Oct4 DNA-bound active fraction was studied in mesendoderm of *nanog* morphants at the same stage (oblong; 3.5 hpf). A low dose of *nanog* MO, which did not cause severe developmental defects, was co-injected with *GFP-Oct4* mRNA. This co-injection led to a reduction in the bound fraction of GFP-Oct4 from $0.27 \pm 0.01$ to $0.22 \pm 0.01$ (p<0.001; *Figure 2—figure supplement 1* and *Figure 2—source data 1*). As a control, we carried out the reverse experiment to assess

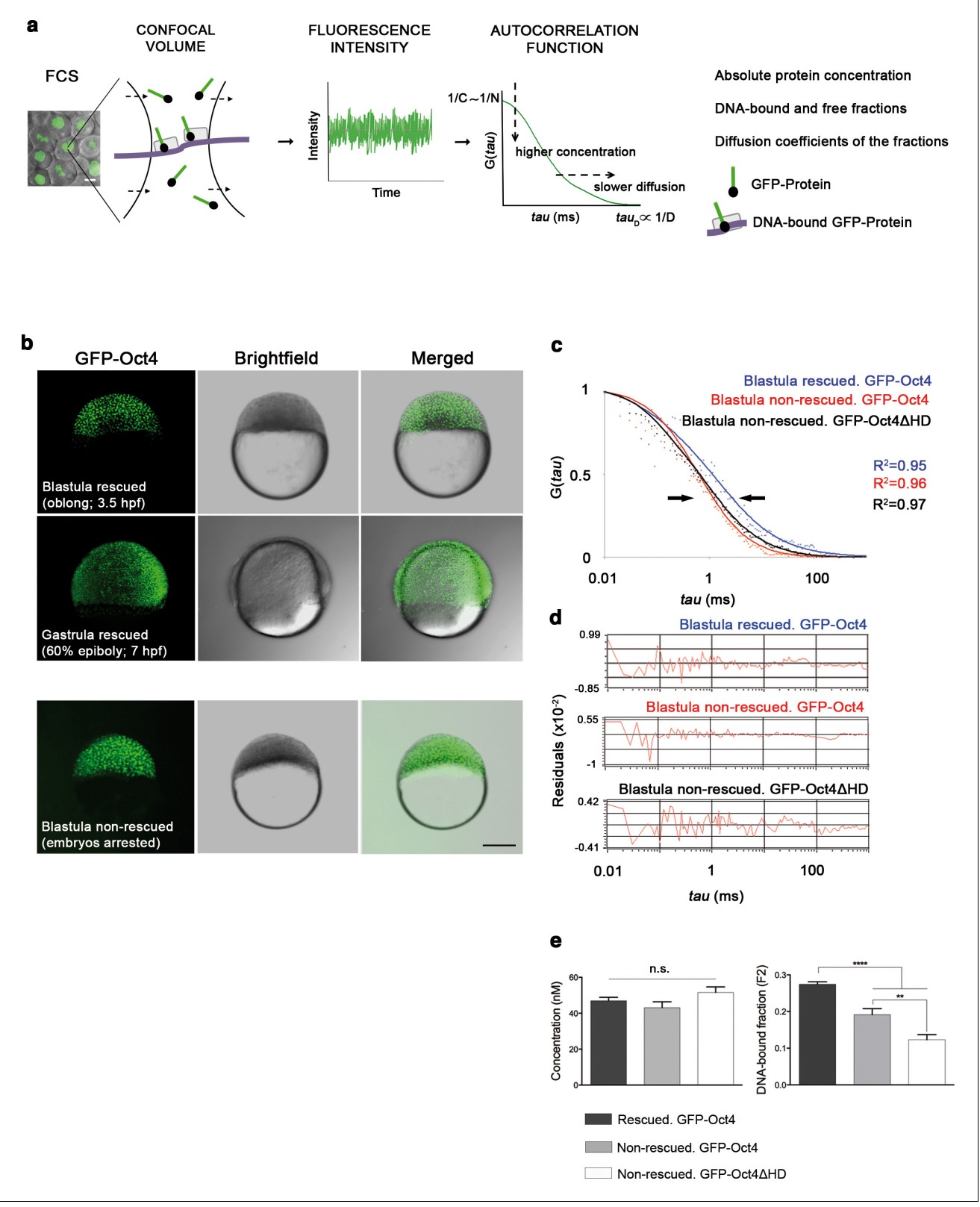

**Figure 1.** Oct4 DNA-bound active fraction controls zebrafish gastrulation. (**a**) Schematic diagram of fluorescence correlation spectroscopy (FCS). GFP-tagged nuclear protein is localized in the nucleus of embryonic cells. Fluorescence molecules diffuse through a confocal volume ($<1 \ \mu m^3$) within a single-cell nucleus and generate fluctuating fluorescence intensity. The autocorrelation function (ACF) of the fluctuation is fit to obtain the absolute protein concentration (C) and the diffusion coefficient (D), where N is the number of molecules. Scale bar: 10 μm. (**b**) Lateral view of *pou5f3* morphant

*Figure 1 continued on next page*

*Figure 1 continued*

embryos expressing GFP-Oct4 rescued by *GFP-Oct4* mRNA at the blastula [3.5 hr post-fertilization (hpf)] and gastrula (7 hpf) stages and non-rescued embryos (arrested) at the blastula stage. The non-rescued embryos also express GFP-Oct4 but remain at the blastula stage and do not develop further. Scale bar: 200 μm. (c) ACF of the intensity traces of GFP-Oct4 and GFP-Oct4ΔHD in rescued and non-rescued embryos at the blastula stage. The ACF were fit by two-component anomalous diffusion model. Curves are normalized to compare differences in protein activity (indicated by arrows). (d) Raw data of residuals from fit curves shown in c. (e) Concentration and DNA-bound fraction levels derived from the FCS measurements in c. Values represent the mean ± SEM of data from three to five independent experiments ($n$ = 39–125 cell nuclei from 10 to 15 embryos ****p<0.0001; **p<0.01). n.s. over bars indicates non-significant differences. See also *Figure 1—figure supplements 1–3*, *Figure 1—source data 1* and Materials and methods.

The following source data and figure supplements are available for figure 1:

**Source data 1.** Quantification of GFP-Oct4 concentration and activity in zebrafish rescued and non-rescued embryos.

**Figure supplement 1.** GFP-Oct4 rescues zebrafish Pou5f3 function.

**Figure supplement 2.** One- and two-component anomalous diffusion model for GFP and GFP-Oct4.

**Figure supplement 3.** Oct4 concentration and DNA-bound active fraction in embryos rescued with different amount of *GFP-Oct4 mRNA*.

the GFP-Nanog DNA-bound active fraction in *pou5f3* morphants, measuring a reduction from 0.21 ± 0.01 to 0.16 ± 0.01 (p<0.001; *Figure 2—figure supplement 3* and *Figure 2—source data 1*). These FCS data suggest that the Pou5f3- and Nanog-binding fractions are dependent on each other in mesendoderm lineage.

Since Oct4 and Nanog share many genomic binding sites (*Loh et al., 2006*; *van den Berg et al., 2010*), we next investigated whether they interact in vivo in a cell lineage-dependent manner. We used FCCS (Fluorescence Cross-Correlation Spectroscopy; *Bacia et al., 2006*; *Krieger et al., 2015*) to study the diffusion of both proteins simultaneously in mesendoderm and ectoderm at the oblong stage (3.5 hpf; *Figure 2—figure supplement 4* and *Figure 2—source data 2*). The cross-correlation function indicates whether GFP-Oct4 and mCherry-Nanog diffuse together within the same complex, and by calculating the dissociation protein constants (*Kd*), we can then determine the binding affinity of GFP-Oct4 and mCherry-Nanog. We measured a *Kd* of 15.34 ± 1.6 nM and 61.9 ± 7.5 nM in the mesendoderm and ectoderm, respectively, suggesting higher binding affinity in mesendoderm at the oblong stage (*Figure 2—figure supplement 4* and *Figure 2—source data 2*).

To confirm the interaction of Oct4 and Nanog in a cell-lineage dependent manner, we next used FLIM-FRET (Fluorescence Lifetime Imaging Microscopy–Förster Resonance Energy Transfer) in single cells of the developing embryo. FLIM follows the lifetime of the excited state of fluorescent molecules and can be used to monitor protein–protein interactions via FRET (*Sun et al., 2011*; *Margnieanu et al., 2016*) (*Figure 2b*). The proximity of a GFP fusion protein as a donor and an mCherry fusion protein as an acceptor reduces the lifetime of the donor, indicating a protein–protein interaction. Contrarily, no change in the lifetime indicates that the proteins are not interacting (*Figure 2b*). We have previously shown that Oct4 interacts with Sox2 in ESCs and iPSCs using this method (*Lam et al., 2012*). Here, we used a GFP-mCherry tandem protein as a positive control to document the decrease in GFP lifetime when it interacts with mCherry as compared to the lifetime of GFP alone in zebrafish embryos (p<0.0001; *Figure 2c*). *mCherry-Nanog* mRNA, co-injected with *GFP-Oct4ΔHD mRNA* was used as negative control, because the DNA binding domain of Oct4 is needed for its interaction with Nanog and, therefore, the GFP-Oct4ΔHD lifetime should not change in the presence of mCherry-Nanog (*Figure 2c*).

In the nuclei of individual cells within the mesendoderm and ectoderm of oblong stage embryos (3.5 hpf; *Figure 2d*), we found a significant reduction in the GFP-Oct4 lifetime in the presence of mCherry-Nanog in the mesendoderm (1.79 ± 0.05 ns; p<0.0001) but not in the ectoderm (2.28 ± 0.01 ns; *Figure 2d*) as compared with the GFP-Oct4 lifetime in the absence of mCherry-Nanog (2.30 ± 0.01 ns; *Figure 2d*); the binding percentages (*Orthaus et al., 2009*) in the mesendoderm and ectoderm were 27% and 2.2%, respectively (*Figure 2d*). These FLIM-FRET results demonstrate a higher proportion of Oct4–Nanog complexes in the mesendoderm as compared with that in

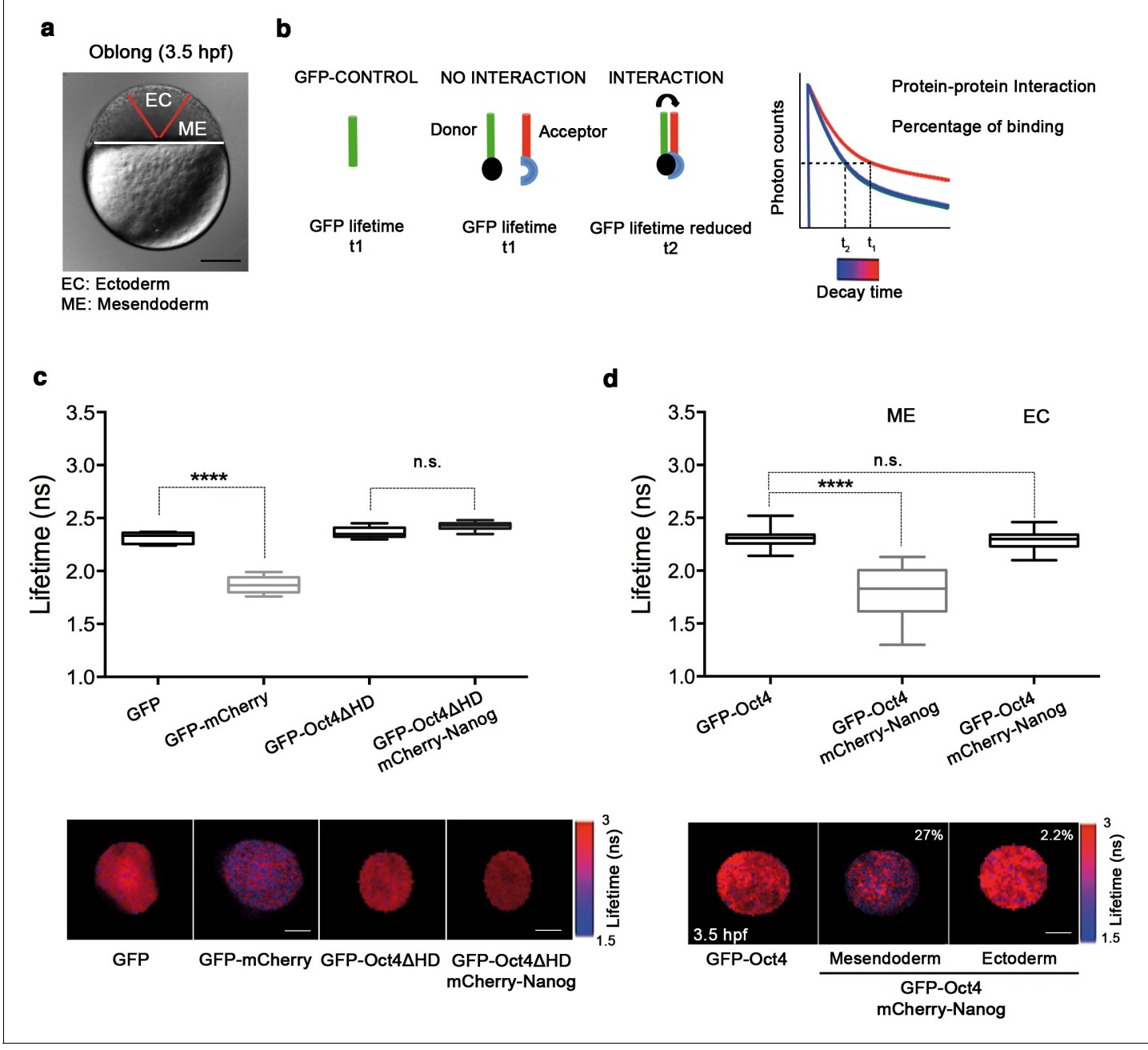

**Figure 2.** Oct4 and Nanog bind in mesendoderm of zebrafish blastula embryos. (**a**) Schematic location of the presumptive mesendoderm (ME) and ectoderm (EC) in an embryo at oblong stage [3.5 hr post-fertilization (hpf)]. Scale bar: 200 μm. (**b**) Schematic diagram of FLIM-FRET (Fluorescence Lifetime Microscopy–Forster Resonance Energy Transfer). GFP lifetime (t1) of the donor is reduced if an acceptor (mCherry) is in close proximity (1–10 nm); this is the reduced lifetime (t2). If the acceptor is not in close proximity (>10 nm) to the donor, donor lifetime remains unchanged (t2 similar to t1). Lifetimes are measured with time-correlated single-photon counting. (**c**) Lifetime values and FLIM images of GFP-Oct4 alone and in the presence of a linked mCherry protein in single cells. Scale bar: 10 μm. In the same graph, lifetime values and FLIM images of GFP-Oct4ΔHD alone and co-expressing mCherry-Nanog in single nuclei. Scale bar: 5 μm. (**d**) Lifetime values and FLIM images of GFP-Oct4 lifetime alone and in the presence of mCherry-Nanog in the nucleus of individual cells from mesendoderm or ectoderm. Scale bar: 5 μm. The percentage of binding is indicated at the top right corner of the FLIM images. Values represent the median and quartile ranges of data from three to five independent experiments (*n* = 20–40 cell nuclei from 10 to 15 embryos; ****p<0.0001). n.s. over bars indicates non-significant differences. See also *Figure 2—figure supplements 1–4*, *Figure 2—source data 1* and *2*.

The following source data and figure supplements are available for figure 2:

**Source data 1.** Quantification of GFP-Oct4 and GFP-Nanog activity in mesendoderm and ectoderm of *wt* and morphant zebrafish embryos.

*Figure 2 continued on next page*

*Figure 2 continued*

**Source data 2.** FCCS parameters of GFP-Nanog and mCherry-Oct4 in mesendoderm and ectoderm of blastula embryos (oblong stage; 3.5 hpf).
**Figure supplement 1.** GFP-Oct4 dynamics in blastula embryos.
**Figure supplement 2.** Dextran red does not interfere in the FCS measurements.
**Figure supplement 3.** GFP-Nanog dynamics in blastula embryos.
**Figure supplement 4.** Oct4 and Nanog cross-correlate in mesendoderm of blastula embryos.

ectoderm at the oblong stage and suggest that Pou5f3 and Nanog co-regulate mesendoderm targets during gastrulation.

## Pou5f3–Nanog complexes are restricted to ventrolateral mesendoderm

Given the proportion of complexes in the oblong stage, we next sought to investigate the spatial localization of Oct4–Nanog complexes in ventral, lateral and dorsal ectoderm and mesendoderm at the 50% epiboly stage (germ ring; 5.7 hpf). This time we used FCCS to explore the diffusion of GFP-Nanog and mCherry-Oct4 and their binding affinities in the different ectodermal and mesendodermal areas. At 50% epiboly, the high $Kd$ value indicated a low-binding affinity between Oct4 and Nanog in the ventrolateral ectodermal cells ($Kd$: 57.09 ± 7.06 nM), and almost a complete absence of cross-correlation in the dorsal ectoderm ($Kd$: 201 ± 49.4 nM) (*Figure 3—figure supplement 1* and *Figure 3—source data 1*). In contrast, the $Kd$ values in the ventral and lateral mesendoderm (5.4 ± 0.4 nM and 11.1 ± 0.73 nM, respectively) indicated a high-binding affinity in these regions as compared with that in the dorsal mesendoderm (36.8 ± 3.5 nM) (*Figure 3—figure supplement 2*) and *Figure 3—source data 1*). We further confirmed these mesendodermal interactions using FLIM-FRET. We found that the GFP-Oct4 lifetime was significantly reduced in the presence of mCherry-Nanog in the ventral and lateral mesendoderm (2.29 ± 0.01 ns in absence of mCherry-Nanog to 2.02 ± 0.01 ns and 2.03 ± 0.02 ns, respectively; p<0.0001; *Figure 3b*) but not in dorsal mesendoderm (2.31 ± 0.01 ns; *Figure 3b*). The percentage of binding was greater than 20% in the ventral and lateral mesendoderm but only 4.7% in the dorsal area (*Figure 3b*).

Finally, we used LiCl treatment, which activates Wnt/β-catenin signaling (*Shao et al., 2012*), to assess the distribution of Oct4–Nanog complexes in dorsalized embryos. Embryos at the 32-cell stage were treated with LiCl and monitored for changes in Oct4–Nanog complexes using FLIM-FRET. As expected, LiCl-treated embryos exhibited radialized dorsal structures along the germ ring (*Figure 3b*). The cells from these radial dorsal structures showed a similar percentage of Oct4–Nanog binding to that of cells from the dorsal mesendoderm of non-treated embryos (2% and 4.7%, respectively; *Figure 3b*). These results suggest that uneven yet specific levels of the Pou5f3–Nanog complex may drive DV mesendodermal patterning.

## Nanog cooperates with Pou5f3 to promote ventral fate

Pou5f3 is known to act in DV patterning (*Reim and Brand, 2006*). To address whether Nanog is involved with Pou5f3 in driving DV patterning, we performed a series of Nanog loss-of-function experiments using varying amounts of *nanog* MO (*Figure 4—figure supplement 1*). With 1.6 ng of *nanog* MO, we observed that 90% of morphants were Class (C) IV (poorly developed axial structures, and no apparent eyes, trunk or tail) (*Figure 4a,b*); with 0.4 ng of *nanog* MO, 70% of morphants showed C III (truncated bodies and hypoplastic eyes) and C II phenotypes (normal head structures, a short body axis, a bent tail and no fins) (*Figure 4a,b*). Near complete rescue (C I) was observed when 0.1 ng of *nanog* mRNA (lacking the MO-target site) was co-injected with 0.4 ng of MO (*Figure 4a,b*). Embryos injected with *nanog* mismatch-MO (5 mismatch nucleotides; *nanog\**) developed normally (data not shown).

We further analyzed the specification of the DV axis, which is driven by a gradient of BMP activity, by measuring the expression of specific markers. In particular, the expression of *chd* at the dorsal

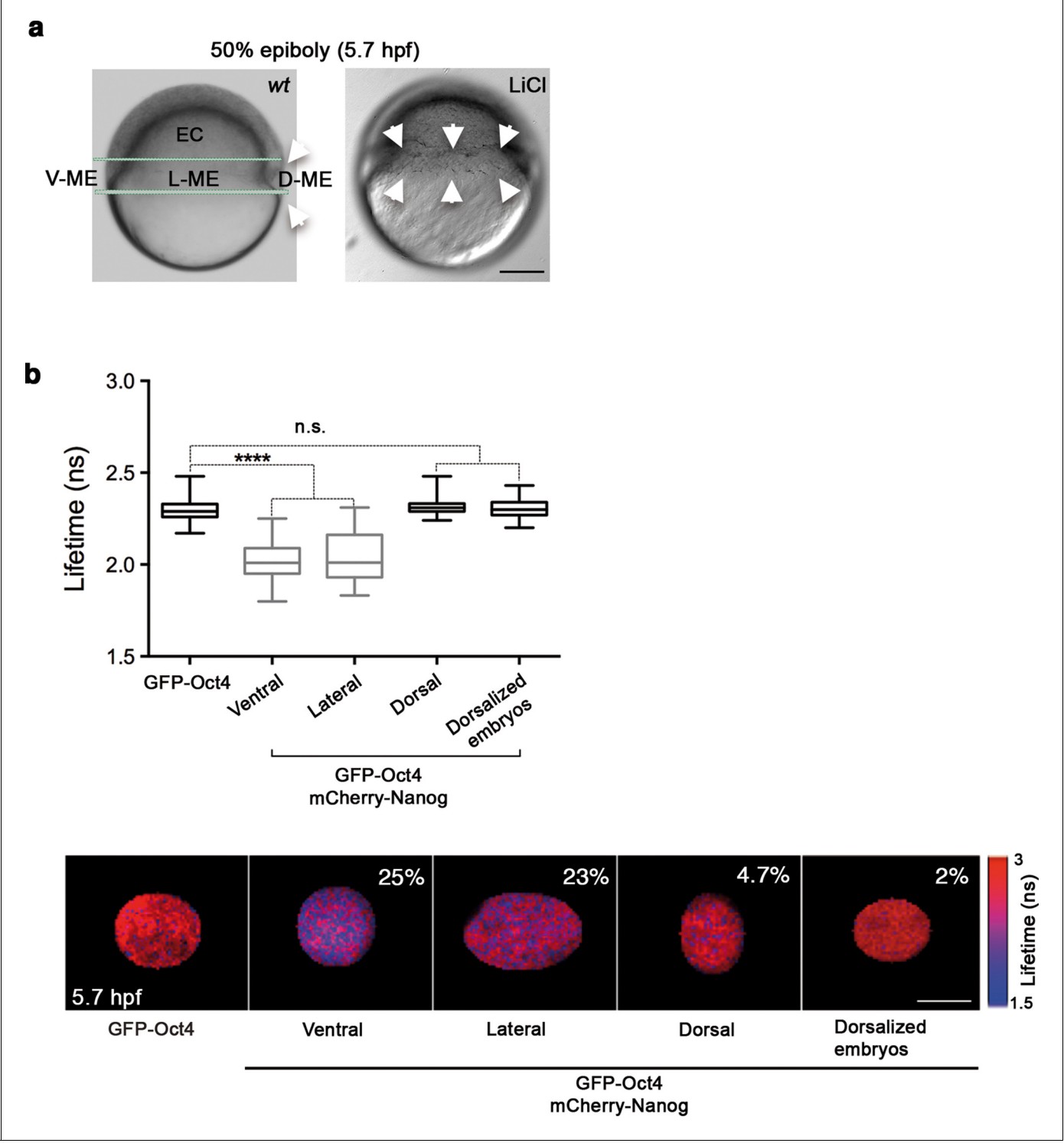

**Figure 3.** Oct4 and Nanog complexes in the ventrolateral mesendoderm. (**a**) *wt* embryo showing the ventral (V)–lateral (L)- and dorsal (D)-mesendoderm (ME) at the 50% epiboly [5.7 hr post-fertilization (hpf)] before commencement of involution. LiCl-dorsalized embryo at 5.7 hpf. Arrows show the dorsal organizer of the *wt* embryo and radialized dorsal structures along the germ ring of dorsalized embryos. Scale bar: 200 µm (**b**) Lifetime values and FLIM images of GFP-Oct4 lifetime alone and in the presence of mCherry-Nanog in the nuclei of individual cells measured at different locations within the mesendoderm. Values represent the median and quartile ranges of data from three to five independent experiments (*n* = 30–70 cell nuclei from 10 to 15 embryos; ****p<0.0001). The percentage of binding is indicated at the top right corner of the FLIM images. Scale bar: 5 µm. n.s. over bars indicates non-significant differences. See also *Figure 3—figure supplements 1,2* and *Figure 3—source data 1*.

*Figure 3 continued on next page*

*Figure 3 continued*

The following source data and figure supplements are available for figure 3:

**Source data 1.** FCCS parameters of GFP-Nanog and mCherry-Oct4 in mesendoderm and ectoderm of gastrula embryos (50% epiboly; 5.7 hpf).
**Figure supplement 1.** Nanog and Oct4 cross-correlation in the ectoderm of gastrula embryos.
**Figure supplement 2.** Nanog and Oct4 cross-correlation in ventrolateral mesendoderm of gastrula embryos.

pole was up-regulated dramatically: at the 30% epiboly stage, *chd* was expressed circumferentially, whereas at 50% epiboly, *chd* transcripts were ubiquitous (*Figure 4c*). *gsc*, a dorsally expressed gene primarily controlled by the Nodal pathway, was also up-regulated but to a lesser extent, suggesting that Nanog largely controls the levels of BMP signaling (*Figure 4c*). Whereas *bmp2b* expression was localized at the embryonic shield and broadly expressed ventrally at the 70% epiboly stage in controls, *nanog* morphants showed significantly down-regulated *bmp4* (data not shown) and *bmp2b* expression (*Figure 4c*). Consistently, the expression of direct targets of BMP signaling, such as *vox* and *vent*, was greatly reduced in *nanog* morphants at the 50% epiboly stage (*Figure 4c*). The spatial distribution of *pou5f3* transcripts in *nanog* morphants during 50% epiboly was not affected, but the level of expression was reduced (*Figure 4c*). Together, these results indicate that *nanog* knockdown leads to the dorsalization of embryos and that maternal *nanog* activity is necessary for ventral cell specification by BMPs.

The dorsalization and loss of endoderm in *nanog* morphants is reminiscent of the phenotype of MZ*spg* mutants, which are completely devoid of Pou5f3 (*Lunde et al., 2004*; *Reim and Brand, 2006*). This supports and reinforces the idea that Pou5f3 and Nanog may participate in the same developmental program. To test this, MZ*spg* mutants and *nanog* morphants were compared with embryos deficient in both MZ*spg* and *nanog* using markers of DV patterning. Expansion of the dorsal marker, *chd*, was more pronounced in MZ*spg/nanog*-deficient embryos as compared with *nanog* morphants or MZ*spg* mutants at the 50% epiboly stage (*Figure 4d*). Conversely, the ventral marker, *vox*, was absent in MZ*spg/nanog*-deficient embryos but only reduced in both *nanog* morphants and MZspg mutants (*Figure 4d*). Lastly, we tested if *nanog* overexpression could compensate for the loss of *pou5f3* or if *pou5f3* could compensate for the depletion of *nanog*. Injection of *nanog* mRNA into MZ*spg* mutants failed to rescue the expression of *chd* (*Figure 4e*). Similarly, injection of *pou5f3* mRNA into *nanog* morphants could not rescue DV patterning (*Figure 4e*). These results suggest that *pou5f3* and *nanog* display overlapping functions during DV patterning but cannot compensate for one another. This is consistent with the notion that Nanog must cooperate with Pou5f3 to promote ventral fate.

## Sox32 modulates Pou5f3–Nanog complexes to specify mesendoderm along the DV axis

The paucity of Oct4–Nanog complexes in the dorsal mesendoderm at 50% epiboly suggested that some other TFs could partner with Oct4 to replace Nanog. Lineage tracing experiments have shown that dorsal endodermal precursors arise from cells in the dorsal mesendoderm near the margin before the cells involute to form the hipoblast, and that those situated above are restricted to the mesoderm (*Warga and Nüsslein-Volhard, 1999*). The dorsal side is easily detected by the presence of the 'shield' at 50% epiboly, but the ventrolateral endodermal cells are derived from dispersed precursors located close to the margin, which makes them impossible to distinguish morphologically at this stage. Previously, we have shown that Oct4 and Nanog form complexes in the whole mesendoderm at 3.5 hpf and, by 4 hpf, Sox32 expression starts in the dorsal endodermal cells and extends to ventrolateral endoderm (*Thisse et al., 2001*). Thus, we next tested whether Sox32 could compete with Nanog for Oct4 binding in the prospective dorsal endoderm.

In *sox32* morphants, we saw a significant reduction in the lifetime of GFP-Oct4 in the presence of mCherry-Nanog (2.31 ± 0.01 ns to 2.16 ± 0.03 ns, p<0.0001; *Figure 5a-c*), and an increase in the binding percentage from 1% to 15%, indicating that the Oct4–Nanog complex forms in the absence of Sox32 in the dorsal endoderm precursors at 50% epiboly (*Figure 5a-c*). To verify the existence of

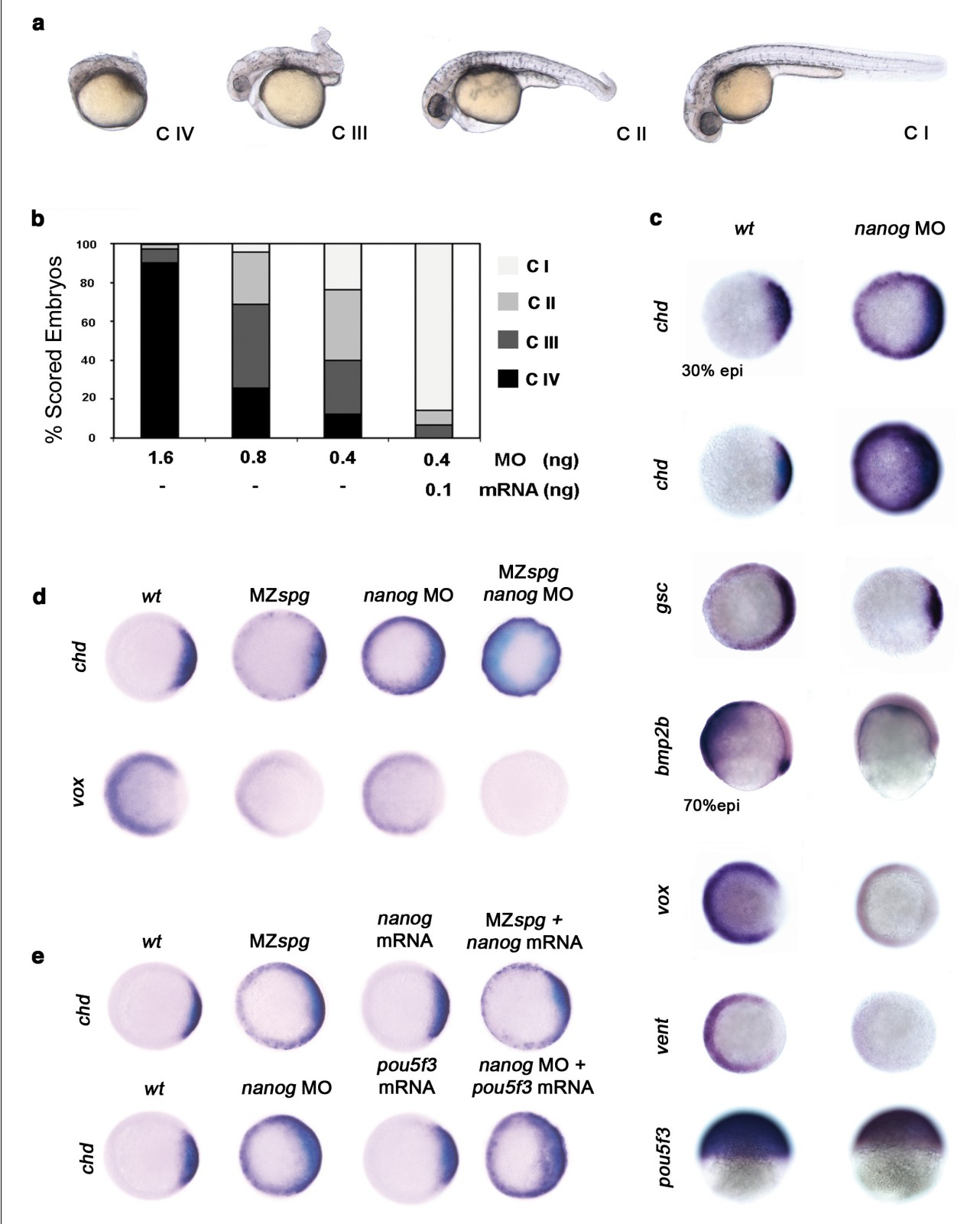

**Figure 4.** Pou5f3 and Nanog promote ventral fate. (a) *nanog* MO-injected larvae show severely affected (Class (C) IV), less-severely affected (C III), mildly affected (C II) and least affected (C I) phenotypes. (b) Relative percentages of C I, C II, C III and C IV larvae according to dose of *nanog* MO injected (1.6 ng of *nanog* MO injection, n = 167; 0.8 ng of *nanog* MO injection, n = 186; 0.4 ng of *nanog* MO injection, n = 143). Co-injection of *nanog* MO (0.4 ng) with *nanog** mRNA (0.1 ng) leads to over 80% *wt*-like larvae as opposed to 20% wt-like larvae in its absence (n = 126). (c–e) Embryos are at

*Figure 4 continued on next page*

*Figure 4 continued*

50%-epiboly except where indicated. Embryos are in top views except lateral views for *bmp2b-*, *oct4-* and *sox17*-stained embryos. Dorsal is to the right-hand side. Markers were analysed following injection of 0.8 ng of *nanog* MO at the 1-cell stage. (c) *chd* expression in the dorsal margin is expanded ventrally in 30%-epiboly *nanog* morphants relative to *wt* embryos (86%, *n* = 40), and is uniformly expressed in the blastoderm of *nanog* morphants at 50%-epiboly relative to *wt* embryos (94%, *n* = 66). *gsc* expression in the prospective shield is expanded ventrally within the germ ring in *nanog* morphants relative to *wt* embryos at the early gastrula stage (71%, *n* = 47). *bmp2b* expression in the ventral ectoderm and organizer is markedly reduced in *nanog* morphants relative to *wt* embryos at mid-gastrulation (96%, *n* = 45). Expression of *vox*, a BMP target, is greatly diminished in *nanog* morphants relative to *wt* embryos (86%, *n* = 68). Expression of *vent* in the ventral margin is nearly absent in *nanog* morphants relative to *wt* embryos (95%, *n* = 56). At the early-gastrula stage, *pou5f3* expression in the blastoderm is reduced in *nanog* morphants compared to *wt* embryos (97%, *n* = 44). (d) Effect of MZ*spg*, *nanog* MO and MZ*spg/nanog* MO on the expression of *chd* and *vox*. *chd* expression in the organizer of *wt* embryos (100%, *n* = 92) is ventrally expanded in MZ*spg* embryos (98%, *n* = 75) and *nanog* morphants (90%, *n* = 97). In *nanog* MO-injected MZ*spg* embryos, *chd* expression is further expanded in the entire blastoderm (92%, *n* = 62). *vox* expression in the ventral margin of *wt* embryos (100%, *n* = 96) is markedly reduced in MZ*spg* embryos (97%, *n* = 60) and *nanog* morphants (88%, *n* = 84). In *nanog* MO-injected MZ*spg* embryos, *vox* expression is completely lost (94%, *n* = 58). (e) Effect of *oct4* mRNA in *nanog* MO, and *nanog* mRNA in MZ*spg* mutant on *chd* expression. *chd* expression is ventrally expanded in MZ*spg* embryos (98%, *n* = 75) relative to *wt* embryos (100%, *n* = 92). *nanog* mRNA cannot cause *chd* expansion when injected into *wt* embryos (93%, *n* = 65) and cannot rescue *chd* ventral expansion when injected into MZ*spg* embryos (94%, *n* = 52). *chd* expression is ventrally expanded in *nanog* morphants (90%, *n* = 97) relative to *wt* embryos (100%, *n* = 92). *Pou5f3* mRNA cannot cause *chd* expansion when injected into *wt* embryos (92%, *n* = 67) and cannot rescue *chd* ventral expansion caused by *nanog* depletion when co-injected with *nanog* MO (88%, *n* = 64). Data are from three to five independent experiments (*n* = 40–150). See also *Figure 4—figure supplement 1*.

The following figure supplement is available for figure 4:

**Figure supplement 1.** *Nanog* controls dorsoventral (DV) patterning.

Oct4–Sox32 complexes in these cells in vivo, we followed the GFP-Sox32 lifetime in the presence and absence of mCherry-Oct4. We found that the GFP-Sox32 lifetime was significantly reduced from $2.54 \pm 0.01$ ns in the absence of mCherry-Oct4 to $2.21 \pm 0.04$ ns in presence of mCherry-Oct4, with a binding percentage of 18% (p<0.0001; *Figure 5a-c*). *Figure 5—figure supplement 2* and *Figure 5—source data 1* show that Nanog also interacts with Sox32 in the dorsal endoderm at 50% epiboly. The co-existence of these Oct4–Sox32 and Nanog–Sox32 complexes was further confirmed in ventrolateral endodermal cells at 60% epiboly (7 hpf) using the Tg(*sox17*:GFP-UTRN) line, which reports *sox17* promoter activity by driving *actin*-GFP (*Woo et al., 2012*) (*Figure 5—figure supplement 2* and *Figure 5—figure supplement 1*). Together, these results suggest that Sox32 prevents the formation of Pou5f3–Nanog complexes at the 50% epiboly stage in dorsal aspects and restricts them to the ventrolateral mesendoderm.

To ascertain whether Oct4 and Nanog still bind in ventrolateral mesoderm and endoderm lineages at 60% epiboly (7 hpf; (*Figure 6a–c*), we measured the GFP-Oct4 lifetime in the absence and presence of mCherry-Nanog. We observed a significant reduction in GFP-Oct4 lifetime from $2.31 \pm 0.01$ ns to $2.01 \pm 0.02$ ns and $1.98 \pm 0.02$ ns in the ventrolateral mesoderm and endoderm, respectively (p<0.0001; (*Figure 6a–c*), suggesting that Pou5f3–Nanog complexes may be involved in ventrolateral patterning at later stages.

Elabela mutant (*ela^br13^*) embryos have a significantly lower number of endodermal progenitors that express *sox17*; yet, the total levels of *sox17* are still higher in *ela^br13^* embryos than in control siblings (*Chng et al., 2013*). Likewise, *sox32* levels are higher in *ela^br13^* embryos than in control siblings (p<0.01; *Figure 6d*). To gain insight into this conundrum, we next followed Oct4–Nanog complexes in ventrolateral mesodermal and endodermal cells of Tg (*sox17*:GFP-UTRN; *ela^br13^*) embryos. We observed a significant decrease in the GFP-Oct4 lifetime, suggesting that Pou5f3 and Nanog also interact in the ventrolateral mesoderm and endoderm of *ela^br13^* embryos (*Figure 6b*). Interestingly, in contrast to mesodermal cells, the percentage of Oct4–Nanog complexes in endodermal cells was significantly reduced from 21% to 15% (p<0.01; *Figure 6e*). These results support the idea that Sox32 competes with Nanog for Pou5f3 binding to regulate cell fate.

Next, we tested whether the decrease in the percentage of Oct4–Nanog complexes in ventrolateral endoderm of *ela^br13^* embryos would affect BMP signaling. We found that *bmp4* expression was more reduced and restricted to the ventral area of *ela^br13^* embryos. *bmp2b* and *bmp2a* expression was also lower in *ela^br13^* embryos as compared to *wt* embryos (*Figure 6f*). The expression of ventrolateral mesoderm *bmp* targets, such as *tbx6* and *eve1*, were slightly reduced in *ela^br13^* as compared

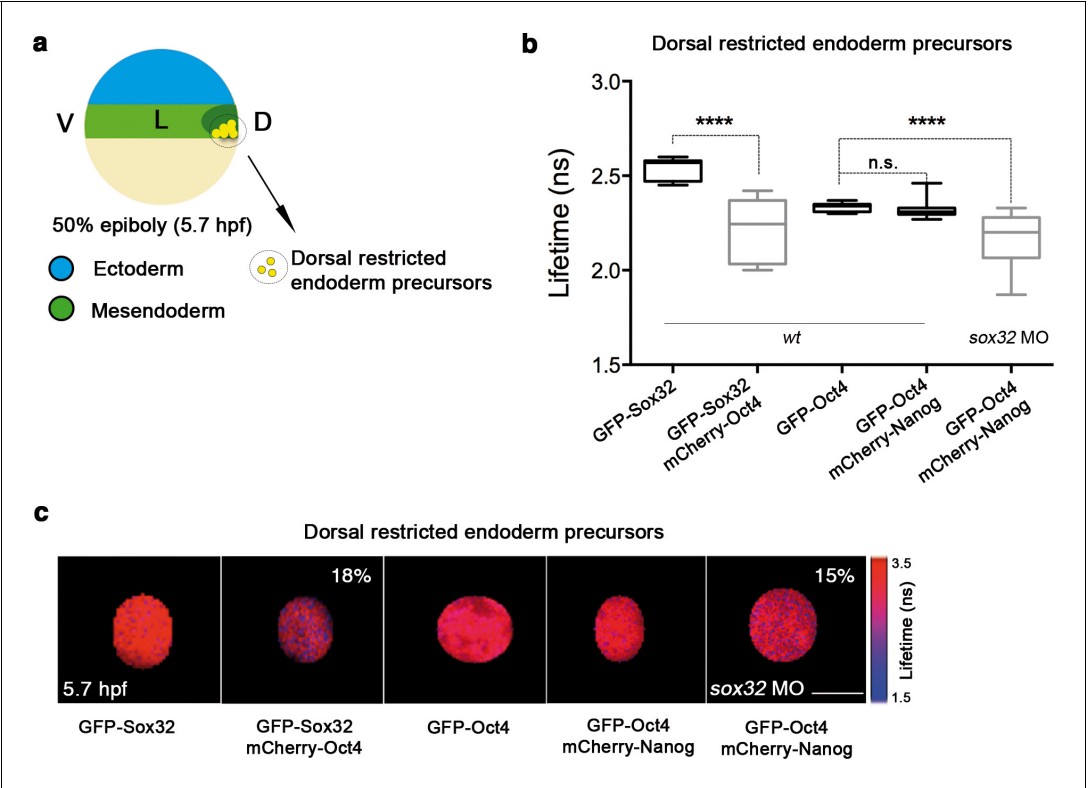

**Figure 5.** Sox32 competes with Nanog for Oct4 binding at dorsal endoderm of gastrula embryos. (**a**) The schematic shows the main germ layers of the embryo at 50% epiboly (5.7 hpf) with ectoderm in blue and mesendoderm in green. The dorsal-restricted endoderm precursors are shown in yellow. (**b**, **c**) Lifetime values (**b**) and FLIM images (**c**) of GFP-Sox32 and GFP-Oct4 alone and in the presence of mCherry-Oct4 and mCherry-Nanog, respectively, in the nuclei of individual cells of dorsal endoderm precursors. Scale bar: 5 μm values represent the median and quartile ranges of data from three to five independent experiments (*n* = 20–30 cell nuclei from 10 embryos; ****p<0.0001). See also *Figure 5—figure supplement 2*, *Figure 5—source data 1*.

The following source data and figure supplements are available for figure 5:

**Source data 1.** FCCS parameters of GFP-Sox32 and mCherry-Nanog in endoderm of gastrula embryos (50% epiboly; 5.7 hpf).

**Figure supplement 1.** Sox32 binds Oct4 in ventrolateral endoderm.

**Figure supplement 2.** Nanog and Sox32 interact in endoderm.

with *wt* embryos, whereas *her5*, which is expressed in a dorsal subpopulation of endodermal precursors (*Müller et al., 1996b*), was significantly increased (*Figure 6f–g*). Dorsal mesoderm markers, such as *gsc* and *chd*, were not significantly upregulated in *ela^br13* embryos relative to *wt* embryos. Ventral and dorsal ectoderm, marked by *gata2* and *otx2*, respectively, were not affected in *ela^br13* mutants (*Figure 6—figure supplement 1*). These results suggest that elevated levels of Sox32, as observed in Elabela mutant embryos, may decrease the proportion of Pou5f3–Nanog complexes in ventrolateral endodermal cells, which, in turn, may lower BMP signaling in the mesendoderm, but not ectoderm, lineage.

## Discussion

Protein dynamics critically regulate tissue differentiation and morphogenesis. Although the dynamics of receptor–ligand interactions (*Ries et al., 2009*; *Shi et al., 2009*) and the activity gradients of morphogens (*Dubrulle et al., 2015*; *Müller et al., 2012*; *Yu et al., 2009*) have been studied in zebrafish and *Xenopus* embryos, very little is known about TF dynamics and the interactions that drive early cell commitment in vivo. In this work, we provide quantitative data for Oct4—and, by

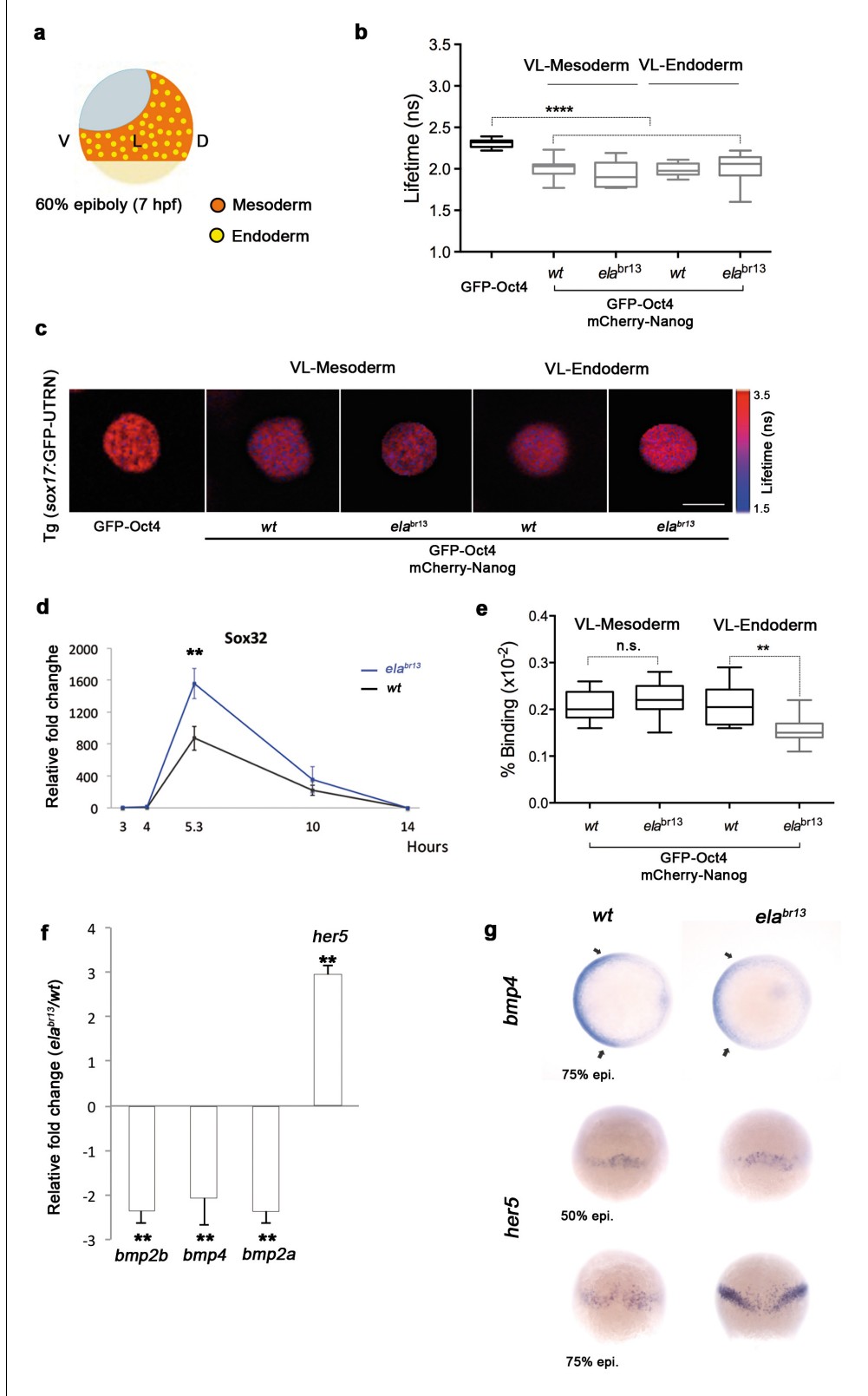

**Figure 6.** Sox32 modulates Oct4 and Nanog complexes in ventrolateral endoderm of gastrula embryos. (a) The schematic shows the Ventrolateral (VL)- and dorsal (D)-mesendoderm layers at 60% epiboly (7 hpf), with mesoderm in orange and endoderm in yellow. (b, c) Lifetime values (b) and FLIM images (c) of GFP-Oct4 alone and in the presence of mCherry-Nanog in the nuclei of individual cells within VL-Mesoderm and VL-Endoderm cells of *wt* and *ela*[br13] mutants. Values of FLIM data represent the median and quartile ranges of data from three to five independent experiments (*n* = 20–40 cell

*Figure 6 continued on next page*

*Figure 6 continued*

nuclei from 10 embryos; ***p<0.0001). Scale bar: 5 μm. (d) qRT-PCR analysis of *sox32* relative to *actin* in *wt* and *ela*[br13] mutant embryos. Values represent mean ± SEM of data from three independent experiments (**p<0.01). (e) Graphs show percentage of binding of GFP-Oct4 and mCherry-Nanog in VL-Mesoderm and VL-Endoderm of *wt* and *ela*[br13] mutant embryos. Values represent the median and quartile ranges from data of three to five independent experiments (*n* = 20–40 cell nuclei from 7 to 10 embryos; **p<0.01). Values represent mean ± SEM of data from three independent experiments (**p<0.01). n.s. over bars indicates non-significant differences. (f) qRT-PCR analysis relative to *actin* reveals different transcription levels of *bmp2b, bmp4, bmp2a* and *her5* in *ela*[br13] mutants at 60% epiboly (7 hpf). Values represent mean ± SEM of data from three independent experiments (**p<0.01). (g) *bmp4* expression (top view, dorsal is to the right-hand side) is ventrally reduced in *ela*[br13] mutants compared with *wt* embryos. *her5* expression (dorsal view) is dorsally upregulated in *ela*[br13] mutants related to *wt* embryos. See also *Figure 6—figure supplement 1*.

The following figure supplement is available for figure 6:

**Figure supplement 1.** Ela/Aplnr pathway refines ventrolateral Bmp signaling.

extension, Pou5f3—DNA binding, and the spatiotemporal complexes that form to achieve proper germ layer formation during gastrulation.

Our FCS experiments show that the rescue of maternal *spg* function depends on Oct4 DNA-binding. Others have proposed that TFs output can be influenced by the dynamic changes following cellular stress (*Cai et al., 2008*; *Ashall et al., 2009*; *Purvis et al., 2012*: *Petrenko et al., 2013*). In particular, chromatin modifications can influence long- and short-lived TF DNA binding in early mouse embryos (*White et al, 2016*). Hence, it is possible that the cellular stress inherent to our rescue assay may, in part, account for the decrease in the Oct4 DNA-bound fraction in our non-rescued embryos. Because we are able to document differences in the distribution of the Oct4 DNA-bound fraction between distinct cell lineages, with higher fractions measured in mesendoderm versus ectoderm, our results cannot be explained by a non-specific stochastic rescue. DNA accessibility is thought to impact the degree to which Oct4 and Sox2 are able to find their target binding sites and interact (*Lam et al., 2012*). Our results, showing a lineage dependency on such an Oct4–DNA interaction, support the view that a switch in the partners of Pou5f3 must play a role in cell fate determination.

Using FLIM, we provide the first evidence for complex formation between Oct4 and other TFs in developing embryos. Nanog, which activates zygotic transcription with Pou5f3 and SoxB1 (*Lee et al., 2013*; *Leichsenring et al., 2013*), formed a significantly higher percentage of complexes with Oct4 in the mesendoderm as compared with the ectoderm at the blastula stage. Given that Sox proteins in mesendoderm do not co-regulatePou5f3 targets (*Onichtchouk et al., 2010*), we speculate that Pou5f3 and Nanog may act together to control zygotic genes in mesendoderm and thereby limit lineage commitment in pluripotent cells. At the gastrula stage, our results show that Nanog cooperates with Pou5f3 in ventrolateral mesendoderm to promote ventral fate by acting upstream of BMP signaling. BMP signaling is initially uniform at the blastula stage and is shaped by inhibitors from the dorsal side at the beginning of gastrulation (*Blader et al., 1997*; *Kishimoto et al., 1997*; *Reversade and De Robertis, 2005*). The presence of Oct4–Nanog complexes at the ventrolateral but not dorsal mesendoderm suggests that, during the transition from blastula to gastrula, dorsal factors may compete with Nanog to partner with Pou5f3, which in turn, may restrict the expression of BMP ligands to the ventral aspects of the embryo. In dorsal endoderm precursors, Oct4–Nanog complexes are detectable following the knock down of *sox32*, indicating that Sox32, which interacts in vivo with Oct4 in these cells, competes with Nanog for Oct4 binding. We also found that Nanog complexes with Sox32 in the same group of cells, suggesting that Sox32 prevents the formation of Pou5f3–Nanog complexes in dorsal endoderm precursors by binding either with Pou5f3 or Nanog. The presence of the Sox32–Nanog complex suggests that Nanog, in addition to its role with Pou5f3 upstream of BMP signaling and in regulating ventrolateral endoderm through the Nodal pathway (*Xu et al., 2012*), may have other roles in patterning the DV axis in zebrafish.

We also provide evidence that Oct4–Nanog, Sox32–Oct4 and Sox32–Nanog complexes coexist in ventrolateral endodermal cells during gastrulation. The Elabela / Aplnr pathway is essential for proper endoderm differentiation, as it regulates the proliferation and migration of the endoderm precursors (www.elabela.com; *Chng et al., 2013*; *Pauli et al., 2014*). Interestingly, when the Ela pathway is inhibited, the levels of *sox32* increase, accompanied by a decrease in the percentage of

Oct4–Nanog complexes in endodermal cells. Thus, these results support the idea that Sox32 regulates Pou5f3–Nanog complexes by protein competition and, thus, modulates endoderm formation in a spatiotemporal manner along the DV axis. The modification of these Pou5f3–Nanog complexes in ventrolateral endodermal cells of Ela-null embryos alters the expression of BMP targets, such as *her5, eve1, tbx6* and *her5*, in particular, is known to control the anteroposterior migration of endodermal progenitors (*Tiso et al., 2002*), a process that is acutely affected in Ela-null embryos. We did not detect significant changes in the expression of the dorsal markers, *gsc* and *chd,* suggesting that the observed decrease in BMP signaling is not sufficient to overtly dorsalize the embryo. Thus, our data show that the Elabela / Aplnr pathway, by controlling *sox32* levels, quantitatively modulates Oct4-Nanog complex formation to refine BMP signaling during DV patterning of the endoderm. Overall, our in vivo measurements of TF complex formation and our understanding of the protein competition among Pou5f3, Nanog and Sox32 provide a quantitative overview of how key TFs control the upstream transcription of important morphogens, including the BMP pathway, that initiate histotopic differentiation along the three embryonic axis.

## Material and methods

### Zebrafish strains and husbandry
Adult zebrafish of the *wt* (AB) strain were kept and bred under standard conditions at 28.5°C (*Westerfield, 1993*). The *spg^{m793-/+}* and Tg(*sox17*:GFP-UTRN) strains were obtained from the laboratory of Drs. Wolfgang Driever (University of Freiburg, Germany) and Didier Stainier (Max Planck Institute, Germany), respectively. To generate embryos that were both maternal and zygotic mutants for *pou5f3*, mutant adult carriers (MZ*spg*) were generated using zebrafish *pou5f3* mRNA-mediated rescue of *spg^{m793-/-}* embryos, as described previously (*Reim and Brand, 2006*). *ela^{br13}* mutants have been previously described (*Chng et al., 2013*). Embryos were collected by natural spawning and staged as described elsewhere (*Kimmel et al., 1995*; *Westerfield, 1993*).

### Cloning and fusion proteins
Zebrafish *nanog* and *pou5f3* were cloned into pCS2+ with the following primers: *nanog* forward: 5'-GTTTATCTAACGGCGAAATGGCG-3', *nanog* reverse: 5'GCAACCCATGACATCACTGCCT-3', *pou5f3* forward: 5'-ATGACGGAGAGAGCGCAGA-3', and *pou5f3* reverse: 5'-TTAGCTGGTGAGATGACCCAC-3'. To generate a zebrafish *nanog* construct that would be immune to MO translation inhibition (*nanog**), the following forward primer was used: 5'-GGCACCATGGCAGATTGGAAAATGCCGGTG-3'.

GFP, GFP-mCherry, GFP-Oct4, GFP-Oct4ΔHD and mCherry-Oct4 were synthetized as previously described (*Lam et al., 2012*; *Plachta et al., 2011*). To generate GFP-Nanog and mCherry-Nanog fusion proteins, cDNA was amplified using sequence-specific primers and cloned directionally into BamHI and NotI sites of the pXJ40:GFP and pXJ40:mCherry expression vectors, respectively. Sox32 and Vox were provided by Dr. Frederic M. Rosa (Institute National de la Santé et de la Recherche Médicale, France), amplified with specific primers, and cloned into BamHI and XhoI sites of pXJ40: GFP. Capped mRNAs were synthesized with the SP6 or T7 mMessage mMachine Kit (Ambion Inc., Thermo Fisher Scientific, Austin, TX).

### Morpholinos
The antisense morpholino oligonucleotides (MO) were manufactured by Gene Tools: *nanog* MO: 5'-CTGGCATCTTCCAGTCCGCCATTTC-3' and *nanog* 5-mismatch MO: 5'- CTGcCATgTTgCAcTCCcC-CATTTC -3'. *pou5f3* and *sox32* MOs have been previously described (*Burgess et al., 2002*; *Sakaguchi et al., 2001*).

### Microinjections
MO and mRNAs were injected at the one-cell stage in doses as indicated in the Results. For rescue experiments, mRNAs devoid of the MO binding site were co-injected with the MO. Dextran red (200 pl) was injected into two of the four central cells of 16-cell stage embryos.

## Dorsalization

Chorionated embryos at the 32-cell stage were treated with 0.3 M LiCl solution for 8 min after mRNA injection at one-cell stage (*Shao et al., 2012*).

## Developmental RT-PCR and qRT-PCR analysis

Total RNA was extracted from a group of 10 embryos at the designated stage using Trizol reagent (Sigma-Aldrich, St Louis, MO). cDNAs were synthesized from 1 μg of total RNA using random hexamers (Promega, Madison, WI) and Superscript III reverse transcriptase (Invitrogen, Life Technologies, Carlsbad, CA). RT-PCR was performed for 20 cycles and zebrafish *actin* was used as a loading control. qRT-PCR, normalized to *actin* expression levels, was performed with SYBR Green Master Mix (Applied Biosystems, Foster City, CA). The reactions were carried out in triplicate for each experiment, and data are expressed as the mean ± SEM. Significant differences were considered to be those with a p value of *<0.05 and **<0.01. All qRT-PCR primers are listed in the *Supplementary file 1*.

## Whole-mount in situ hybridization (WISH) of zebrafish embryos

The following clones were used to prepare antisense probes for in situ hybridization: *gata2* (*Detrich et al., 1995*), *dlx3* (*Akimenko et al., 1994*), *otx2* (*Mori et al., 1994*), *myoD* (*Weinberg et al., 1996*), *ntl* (*Schulte-Merker et al., 1994*), *sox17* (*Alexander and Stainier, 1999*), *sqt* (*Feldman et al., 1998*), *her1* (*Müller et al., 1996a*), *pou5f3* (*Takeda et al., 1994*), *chd* (*Miller-Bertoglio et al., 1997*), *gsc* (*Thisse et al., 1994*), *bmp2b* (*Martinez-Barbera et al., 1997*), *bmp4* (*Martinez-Barbera et al., 1997*), *vox* (*Melby et al., 2000*), *vent* (*Melby et al., 2000*), *her5* (*Tiso et al., 2002*) and *tbx6* (*Hug et al., 1997*). The *nanog* probe was generated by digestion of the *nanog* open reading frame in pCS2+ with ClaI and transcription using T7 polymerase. WISH was carried out as previously described (*Tian et al., 2010*) (refer to http://www.reversade.com-a.googlepages.com/protocols for detailed protocols). Images were viewed using the Zeiss Axioplan microscope and captured with the Zeiss AxioCam HRc camera (Zeiss, Oberkochen, Germany).

## Live embryo mounting and fluorescence images

Embryos were mounted at specified stages and orientations in 35-mm glass-bottomed petri dishes using 0.8% low-melting agarose and covered with egg water. The fluorescence images were obtained using an Olympus FV-1000 confocal microscope (Olympus, Tokyo, Japan). The excitation light source was a 488- and 559-nm cw laser with a dichroic mirror of 488/559 and respective GFP and mRFP emission filters along with DIC channel images.

## FCS and FCCS acquisition and fitting

FCS experiments were performed on a PicoHarp 300 TCSPC module (PicoQuant, Berlin, Germany) attached to an Olympus FV-1000 confocal microscope (Olympus) with a $60 \times 1.2$ W objective. The excitation light source was a 488-nm cw Ar laser with a 488/559-nm dichroic mirror and a 520/35-nm emission filter. The fluctuating photons in the confocal volume were detected using a SPAD detector. The pinhole was set to 80 μm, corresponding to 0.2 μm back-projected into the focal plane, and the data were acquired using the SymPhoTime 200 software (PicoQuant). Calibration and measurement of the confocal volume is crucial to extract concentrations and diffusion coefficients. This was performed on a day-to-day basis before measurements were taken under identical settings using a solution of 1 nM Atto 488 with a known diffusion coefficient of 400 $\mu m^2 s^{-1}$ at room temperature (*Wachsmuth et al., 2015*). At three measurement points, intensity time traces were recorded for 30 s in the nuclei of embryos expressing either free GFP or Oct4-GFP. A range of 10 to 18 μW laser power was used to minimize photobleaching.

The FCCS acquisition was performed using a setup similar to that used for the FCS measurement, with the exception that an additional laser line 559 cw was used and the mCherry channel was detected using the 615/45 nm filter along with a 560 nm Dichroic mirror to split the green and red channel emission. GFP alone and mCherry alone samples were measured before measuring the experimental samples to correct for cross-talk. The confocal volume for the red channel was measured using Rhodamine 6G.

FCS data analysis followed the previously established workflow (*Wachsmuth et al., 2015*): After calculation of the ACFs and after correction for slow fluctuations, such as photobleaching, the data were fit with a two-component anomalous diffusion model. This very general model converged to effective one-component anomalous diffusion for a real one-component system, such as free GFP. Using the radius and volume of the focus, the diffusion coefficients and concentrations were calculated as previously described (*Wachsmuth et al., 2015*). To ensure that the confocal volume was not affected by aberrations induced by the refractive index mismatch among water, the medium, and the interior of the embryo, we determined the ratio of the mean intensity extracted directly from the time traces and the mean number of molecules in the focus extracted from the fit, referred to as molecular brightness. Whereas the first is very robust, the second is very sensitive to aberrations and decreases in response (*Yu et al., 2009*). Virtually, all FCS measurements revealed a molecular brightness inside a window of ± 10% around the mean value of GFP measured in the most peripheral nuclei; this allowed us to ensure that the resulting concentrations and diffusion coefficients were not biased due to aberrations (*Figure 1—figure supplement 2*).

The free diffusion coefficient (D1) was determined by global fitting: the ACFs were fit with the two-component anomalous diffusion model and the fast component average was taken to fix the free diffusion time value. With the free diffusion time determined, the rest of the parameters were recalculated. Using GFP-Oct4 as an example, global fitting provides a tauD1 of 3030 ± 250 µs; averaging individual fits yields a similar tauD1 of 3000 ± 390 µs (*Figure 1—figure supplement 2*).

*Kd* values and percentage of association were calculated as previously described (*Shi et al., 2009* and *Sudhaharan et al., 2009*).

## FLIM measurement

Time domain FLIM experiments were performed on a Time Correlated Single Photon Counting (TCSPC) system (PicoQuant) attached to an Olympus FV-1000 confocal microscope (Olympus) with a 60 × 1.2 W objective. The excitation light source was a 485-nm pulsed diode laser controlled by a Sepia II (PicoQuant) driver with a dichroic mirror of 488/559 and a 520/30 emission filter. Individual photon arrivals were detected using a SPAD detector, and events were recorded by a PicoHarp 300 TCSPC module. Lifetime analysis was carried out using SymPhoTime 200 software. Mono- and bi-exponential fittings were applied. The percentage of binding was calculated from the amplitudes derived from the bi-exponential fitting, as previously shown (*Orthaus et al., 2009*).

## Statistics

Statistical analyses and graphs were generated using GraphPad Prism, version 6.0. Unpaired, two-tailed Student's t-tests with Welch correction were performed if data passed the normality assumptions; if data did not pass the normality test, it was analysed by the Mann–Whitney method. Median and interquartile ranges are graphed as box and whiskers of non-normal data, and bar graphs show the mean and standard error of mean (SEM) for normal data.

## Acknowledgements

We thank Larry Stanton, Nicolas Plachta and Steffanie Oess for constructive feedback on the manuscript and Eric Lam and Srinivas Ransamy for helpful discussions. We are grateful to Wolfgang Driever and Didier Stainier for sharing the *spg* and Tg(*sox17*:GFP-UTRN) fishes, Frederic M. Rosa for the *sox32* construct and Karuna Sampath for numerous molecular markers used in this study. This work was supported by the Agency for Science, Research and Technology (A*STAR), Singapore.

## Additional information

### Funding

| Funder | Author |
| --- | --- |
| Agency for Science, Technology and Research | Vladimir Korzh Sohail Ahmed Bruno Reversade |

The funder had no role in study design, data collection and interpretation, or the decision to submit the work for publication.

## Author contributions

MP-C, Conceived, designed and performed the experiments, Analyzed data, Prepared figures, Wrote the manuscript; JT, Performed nanog knockdown analysis and WISH experiments; SCC, Performed WISH experiments; KPS, Made fusion constructs; TS, Gave imaging support; CT, Gave zebrafish work support in rescue experiments; MW, Gave physics analysis support; VK, SA, BR, Conceived the project, Reviewed the manuscript

## Author ORCIDs

Bruno Reversade, http://orcid.org/0000-0002-4070-7997

## Ethics

Animal experimentation: All animal experiments were performed in strict accordance with the relevant laws and institutional guidelines of the Institute of Medical Biology, Singapore. All protocols were approved by the animal ethics committee of Singapore (IACUC#130837 protocol).

# Additional files

## Supplementary files

• Supplementary file 1. List of QRT-PCR primers. Forward and reverse primers used in the qRT-PCR experiments. See *Figure 6* and *Figure 4—figure supplement 1*.

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
