## [Decision Letter]

Thank you for submitting your work entitled "Quantitative imaging reveals real-time Pou5f3-Nanog complexes driving dorsoventral mesendoderm patterning in zebrafish" for consideration by *eLife*. Your article has been reviewed by two peer reviewers, and the evaluation has been overseen by Janet Rossant as the Senior Editor.

We apologize for the extensive review time, but the reviewers found the paper complex and somewhat difficult to evaluate. After consultation, the reviewers and the editor agreed that the application of the novel imaging approach and the biological findings were potentially of wide interest and impact on the field. The reviewers have discussed the reviews with one another and the Reviewing editor has drafted this decision to help you prepare a revised submission.

Summary:

This is a timely manuscript in which the authors use a combination of state-of-the-art in vivo imaging Fluorescence Correlation Spectroscopy (FCS) and Fluorescence Lifetime Imaging Microscopy – Förster Resonance Energy Transfer (FLIM-FRET) and morpholino-based combinatorial gene knockdowns to elucidate the dynamic physical and functional interaction of the transcription factors Pou5f3 (Oct4), Nanog and Sox32 during early patterning of the zebrafish embryo. The zebrafish orthologues of the mammalian pluripotency regulators Oct4 and Nanog have been previously shown to be required for zygotic gene activation (Lee et al., 2013; Leichsenring et al., 2013) and for endoderm formation and epiboly (Lunde et al., 2004; Reim et al., 2004; Xu et al., 2012) in the zebrafish. In addition, Pou5f3 has been shown to be required for ventral specification upstream of BMPs during early dorsoventral patterning of the zebrafish embryo (Reim and Brand, 2006), while a corresponding function of Nanog had not been investigated as yet. The main conclusions of the current paper are: 1) Pou5f3 complexes with Nanog to induce mesendoderm 2) in the prospective dorsal endoderm, Sox32 competes with Nanog for Pou5f3 binding, restricting Pou5f3-Nanog complexes to the ventrolateral region.

This is a potentially interesting paper using technically challenging approaches. But the paper itself needs to be more clearly written and explained to the non-specialist and there are some experimental gaps that need to be filled to support the conclusions. Below we outline three specific areas that we believe would benefit from further experiments. The first two are considered essential, the third is desirable, if feasible in a short time window.

Essential points:

1) According to Table 1, FCS can also be performed at gastrula stages. If so, it would be interesting to compare dorsal versus ventrolateral regions at such gastrula stages also via FCS (in addition to FLIM-FRET). To validate one of the main statements of the paper, namely, that Sox32 competes with Nanog for Oct4 partnering exclusively at the dorsal side, one should include the important control: to show, that Oct4/Sox32 complexes are absent (if really so) in ventral and lateral locations of mesendoderm.

2) If Pou5f3 -Nanog complexes are the main driving force of ventral mesoderm, one should not expect an additive effect in MZ*spg* null mutant. A plausible explanation for the observed phenotypes would be that Nanog has an independent role in mesendoderm patterning, by complexing with Sox32 or other main mesendoderm transcription factors. Minimally, Nanog-Sox32 complex formation should be assessed in the dorsal and ventrolateral regions of the gastrula using FCS and FLIM-FRET, with Nanog and Sox32 probes.

3) In light of the roles of Pou5f3 and Nanog in dorsoventral patterning of mesoderm and ectoderm, it is disappointing that the authors only study differential Nanog/Oct4 complex formation along the dorsoventral axis of the endoderm. Indeed, it would be interesting to extend the FLIM-FRET analyses to get to know whether Nanog/Oct4 complex formation is also compromised in the dorsal ectoderm and the dorsal mesoderm, compared to their ventrolateral counterparts. If it is possible to perform this experiment easily, please add some data, or explain why it is not feasible.

---

## [Author Response]

*[…] This is a potentially interesting paper using technically challenging approaches. But the paper itself needs to be more clearly written and explained to the non-specialist and there are some experimental gaps that need to be filled to support the conclusions. Below we outline three specific areas that we believe would benefit from further experiments. The first two are considered essential, the third is desirable, if feasible in a short time window.*

We are pleased that the reviewers found our manuscript to be potentially of wide interest and impact on the field. In the revised manuscript, we have fully addressed the reviewers’ comments. We have also more clearly written and explained the manuscript and figures for the benefit of non-specialist readers. We address the specific comments below.

*Essential points:*

*1) According to Table 1, FCS can also be performed at gastrula stages. If so, it would be interesting to compare dorsal versus ventrolateral regions at such gastrula stages also via FCS (in addition to FLIM-FRET).*

We agree that FCS analyses comparing dorsal vs ventrolateral areas would be of interest. Instead of single FCS we have used FCCS (Fluorescence Cross-Correlation Spectroscopy), which allows us to study the diffusion of Oct4 and Nanog proteins simultaneously and their cross-correlation in ventrolateral and dorsal mesendoderm at the gastrula stage (50% epiboly; 5.7 hpf). In our opinion, the cross-correlation analysis strengthens the FLIM results.

Autocorrelation functions (ACFs) provide FCS diffusion parameters whereas cross-correlation functions (CCFs) indicate whether GFP-Nanog and mCherry-Oct4 diffuse together and are associated in the same complex. Calculating the dissociation protein constants (*Kd*), we were able to determine the binding affinity of GFP-Nanog and mCherry-Oct4 in mesendoderm along the DV axis. We also show the percentage of protein association.

Diffusion parameters are summarized in [Supplementary-material SD4-data]. Both Oct4 and Nanog diffuse slower in the dorsal mesendoderm than in the ventrolateral mesendoderm, as indicated by the diffusion coefficient (D_2_); however, these differences were determined to be non-significant.

*Kd* values obtained in the ventral and lateral mesendoderm indicated higher binding affinity in these regions as compared with that of the dorsal area. These results are in concordance with our previous FLIM analysis. ACFs, CCFs and *Kd* plots with protein association values can be found in Figure 3—figure supplement 1.

We have extended this analysis to mesendoderm and ectoderm at the blastula stage (oblong stage; 3.5 hpf).

[Supplementary-material SD3-data] summarizes the diffusion parameters. The FCCS results show that Oct4 diffuses faster than Nanog in both cell lineages, as previously shown by single FCS in [Supplementary-material SD2-data]. No differences were found for the D_2_ values of Oct4 and Nanog between the ectoderm and mesendoderm.

Our results also showed lower *Kd* values in the mesendoderm as compared with the ectoderm, suggesting higher binding affinity of Oct4 and Nanog in mesendoderm as previously shown by FLIM in Figure 2. ACFs, CCFs and *Kd* plots with protein association values can be found in Figure 2—figure supplement 4.

Since this work focuses on the DNA-bound fractions and interactions of different TFs, we have included the tables and raw data related to diffusion coefficients (D_2_) derived by FCCS as supplemental material for readers wanting more details. We believe these changes, along with changes in the way the text has been written, will improve the readability for both specialist and non-specialist readers, and give the specialist readers the opportunity to interrogate the data if desired.

*To validate one of the main statements of the paper, namely, that Sox32 competes with Nanog for Oct4 partnering exclusively at the dorsal side, one should include the important control: to show, that Oct4/Sox32 complexes are absent (if really so) in ventral and lateral locations of mesendoderm.*

We would like to respond by highlighting the matter of Sox32 competing with Nanog for Oct4 binding in the dorsal mesendoderm, as we believe that our explanation was insufficient, and there may be some confusion in the interpretation of our data:

Lineage tracing experiments have shown that dorsal endodermal precursors arise from cells in dorsal mesendoderm near the margin before cells involute to form the hipoblast. Those situated above are restricted to the mesoderm (Warga and Nusslein-Volhard, 1999). The dorsal side is easily detected by the presence of the ‘shield’ at 50% epiboly but the ventrolateral endodermal cells are derived from dispersed precursors located close to the margin, which makes them impossible to distinguish morphologically at this stage. From our previous results at 3.5 hpf, Oct4 and Nanog form complexes in the entire mesendoderm (Figure 2). By 4 hpf, Sox32 expression starts in dorsal endodermal cells and extends to the ventrolateral endoderm (Thisse et al., 2001). Sox32 binds Oct4 in vitro and activate Sox17 in the endoderm (Alexander et al., 1999; Kikuchi et al., 2001; Lunde et al., 2004; Reim et al., 2004). In the present study, when the embryo morphologically shows the dorsal shield, we find that Sox32 replaces Nanog as Oct4 partner in the dorsal endoderm (Figure 5). However, Oct4–Nanog complexes coexist with Oct4–Sox32 complexes in the ventrolateral endoderm at 60% epiboly (7 hpf; Figure 6)—it is only at this time point that endodermal cells can be detected in Tg(sox17:GFP-UTRN) embryos (Figure 5—figure supplement 1. In these cells, Sox32 appears to modulate the percentage of Oct4-Nanog complexes through its interactions with these two factors. Together, these findings suggest that Sox32 prevents gradually the formation of Oct4–Nanog complexes in endoderm progenitors along the DV axis.

To show that Oct4 interacts with Sox32 in ventrolateral endoderm at 50% epiboly (5.7 hpf), we performed FLIM-FRET measurements in cells located next to the margin and analysed the lifetime of the GFP-Sox32 protein. The GFP-Sox32 lifetime was reduced in some of the measured cells, suggesting that they are likely to be endoderm progenitors. These results have been included in Figure 5—figure supplement 1.

*2) If Pou5f3 -Nanog complexes are the main driving force of ventral mesoderm, one should not expect an additive effect in MZspg null mutant. A plausible explanation for the observed phenotypes would be that Nanog has an independent role in mesendoderm patterning, by complexing with Sox32 or other main mesendoderm transcription factors. Minimally, Nanog-Sox32 complex formation should be assessed in the dorsal and ventrolateral regions of the gastrula using FCS and FLIM-FRET, with Nanog and Sox32 probes.*

As suggested by the reviewers, we have investigated the interaction of Nanog with Sox32. FCCS and FLIM analyses were performed to measure diffusion parameters of GFP-Sox32 and mCherry-Nanog and their interaction in ventrolateral and dorsal areas.

Diffusion parameters are summarized in [Supplementary-material SD5-data]. We did not find significant differences in the diffusion coefficients of the DNA-bound fractions (D_2_) in the ventral vs dorsal areas.

In the dorsal endoderm at the 50% epiboly stage (5.7 hpf), the *Kd* value obtained by FCCS analysis indicated a high binding affinity between Nanog and Sox32. FLIM-FRET measurements confirmed the interaction, with the GFP-Sox32 lifetime shown to decrease in the presence of mCherry-Nanog in those cells.

In ventrolateral endodermal cells of Tg(*sox17*:GFP-UTRN) embryos at the 60% epiboly stage (7 hpf), FCCS and FLIM analyses led to similar *Kd* and lifetime values as those measured in the dorsal cells.

Collectively, these findings suggest that Sox32 modulates Oct4–Nanog complexes by binding with either Oct4 or Nanog. The presence of the Sox32–Nanog complexes suggests that Nanog may have other roles in patterning the dorsoventral axis in zebrafish, in addition to its role with Oct4 upstream of BMP signalling and in the regulation of the ventrolateral endoderm through the Nodal pathway (Xu et al., 2012).

ACFs, CCFs, *Kd* plots and lifetimes can be found in Figure 5—figure supplement 2.

*3) In light of the roles of Pou5f3 and Nanog in dorsoventral patterning of mesoderm and ectoderm, it is disappointing that the authors only study differential Nanog/Oct4 complex formation along the dorsoventral axis of the endoderm. Indeed, it would be interesting to extend the FLIM-FRET analyses to get to know whether Nanog/Oct4 complex formation is also compromised in the dorsal ectoderm and the dorsal mesoderm, compared to their ventrolateral counterparts. If it is possible to perform this experiment easily, please add some data, or explain why it is not feasible.*

We also agree that it would be interesting to study the potential role of Oct4–Nanog complexes in other cell lineages. Thus, in line with the reviewers’ recommendation and to provide a complete picture of the Oct4–Nanog complexes in the whole embryo, we have now explored their activity in the ventral and dorsal ectoderm at the gastrula stage (50% epiboly; 5.7 hpf) using FCCS.

The diffusion parameters are detailed in [Supplementary-material SD4-data]. As in the mesendoderm, the ectodermal cells showed slower diffusion of the Oct4- and Nanog DNA-bound fraction (D_2_) in the dorsal as compared with ventral areas.

In ventral ectodermal cells, the *Kd* value indicates a low binding affinity for Oct4 and Nanog, whereas, in the dorsal ectoderm, no cross-correlation between the two proteins was found. These results suggest that the formation of Oct4–Nanog complexes may be also compromised in the dorsal ectoderm. ACFs, CCFs and *Kd* plots with protein association values can be found in Figure 3—figure supplement 2.

As the reviewers’ point out, this work primarily focused on the analysis of Oct4–Nanog complex formation in the endoderm; however, this process was also measured in the mesoderm. The results of FLIM measurements, now included in Figure 3 (alongside the new FCCS data), show that, in the ventrolateral mesendoderm, the lifetime of the GFP-Oct4 was reduced in the presence of mCherry-Nanog, indicating an interaction not only in endoderm but also in mesoderm precursor cells. Please note that the results showed a low interaction in the dorsal tissues (dorsal endoderm and mesoderm). We further extended this analysis to include Sox32, as it competes with Nanog for Oct4 binding in the endoderm lineage.